# Purine-Rich Element Binding Protein Alpha, a Nuclear Matrix Protein, Has a Role in Prostate Cancer Progression

**DOI:** 10.3390/ijms25136911

**Published:** 2024-06-24

**Authors:** Takahiro Inoue, Xin Bao, Takumi Kageyama, Yusuke Sugino, Sho Sekito, Shiori Miyachi, Takeshi Sasaki, Robert Getzenberg

**Affiliations:** 1Department of Nephro-Urologic Surgery and Andrology, Mie University Graduate School of Medicine, 2-174, Edobashi, Tsu 514-0001, Japan; 323m02c@m.mie-u.ac.jp (X.B.); kagetaku@med.mie-u.ac.jp (T.K.); y-sugino@med.mie-u.ac.jp (Y.S.); sekitosho@med.mie-u.ac.jp (S.S.); s-miyachi@med.mie-u.ac.jp (S.M.); t-sasaki@med.mie-u.ac.jp (T.S.); 2Dr. Kiran C Patel College of Allopathic Medicine, Nova Southeastern University, Fort Lauderdale, FL 33328, USA; rhgetzenberg@gmail.com

**Keywords:** prostate cancer, nuclear matrix, purine-rich element binding protein, androgen receptor

## Abstract

Solid tumors as well as leukemias and lymphomas show striking changes in nuclear structure including nuclear size and shape, the number and size of nucleoli, and chromatin texture. These alterations have been used in cancer diagnosis and might be related to the altered functional properties of cancer cells. The nuclear matrix (NM) represents the structural composition of the nucleus and consists of nuclear lamins and pore complexes, an internal ribonucleic protein network, and residual nucleoli. In the nuclear microenvironment, the NM is associated with multi-protein complexes, such as basal transcription factors, signaling proteins, histone-modifying factors, and chromatin remodeling machinery directly or indirectly through scaffolding proteins. Therefore, alterations in the composition of NM could result in altered DNA topology and changes in the interaction of various genes, which could then participate in a cascade of the cancer process. Using an androgen-sensitive prostate cancer cell line, LNCaP, and its androgen-independent derivative, LN96, conventional 2D-proteomic analysis of the NM proteins revealed that purine-rich element binding protein alpha (PURα) was detected in the NM proteins and differentially expressed between the cell lines. In this article, we will review the potential role of the molecule in prostate cancer.

## 1. Introduction

Solid tumors as well as leukemias and lymphomas show striking changes in nuclear structure including nuclear size and shape, the number and size of nucleoli, and chromatin texture. These alterations have been used in cancer diagnosis and might be related to the altered functional properties of cancer cells. The modification of nuclear architecture suggests a functional relationship between nuclear organization and gene expression that might related with resistance against various therapeutics [1].

The nuclear matrix (NM) is the fraction of the eukaryotic nucleus insoluble to detergents and high-salt extractions that manifests as a pan-nuclear fiber-granule network, which represents the structural composition of the nucleus, including nuclear lamins and pore complexes, an internal ribonucleic protein network, and residual nucleoli [2,3,4]. In the nuclear microenvironment, the NM is associated with multi-protein complexes, such as basal transcription factors, signaling proteins, histone-modifying factors, and chromatin remodeling machinery directly or indirectly through scaffolding proteins [5]. The NM is thought to provide a structural framework for the spatial organization of nuclear environments, influencing a variety of nuclear processes such as DNA replication and metabolism, transcription, RNA splicing and processing, RNA transport, control of apoptosis, ribosomal biogenesis, steroid hormone activity, and so forth [6].

The primary standard treatment against advanced prostate cancer was androgen-deprivation therapy, either surgical castration or LHRH analogue, but in spite of castration levels of testosterone, certain cells can adapt and resume proliferation mostly with an associated increase in prostate-specific antigen, resulting in castration-resistant prostate cancer (CRPC) [7,8,9]. The molecular mechanisms related to progression to CRPC have been reported and considered to be categorized into either androgen-receptor (AR)-related or AR-unrelated. As for AR-related mechanisms, AR splice variants, the amplification of *AR* genes, mutations in the *AR*, AR stabilization, and enhancement of *AR* transcriptional activity [7,8,9]. Moreover, the intratumoral production of testosterone and dihydrotestosterone and dependency upon alternate AR-related signal transduction pathways have also been implicated as the mechanisms [7,8,9,10]. On the other hand, the AR-unrelated mechanisms include neuroendocrine transdifferentiation, DNA-repair gene alterations such as defects in Breast cancer 1/2 gene (*BRCA1/2*) or *ATM* genes, and *Wnt* pathway alterations [7,8,9]. Neuroendocrine differentiation is a mechanism of resistance to AR-targeted therapy, mediated by lineage plasticity, the process through which AR-dependent adenocarcinoma cells undergo a phenotypic switch to new AR-independent phenotypes, called treatment-emergent neuroendocrine prostate cancer (t-NEPC) [11]. Recent genomic sequencing revealed that an increased incidence of *RB1* loss and *TP53* mutation or deletion, as well as combined alterations, was detected in patients with t-NEPC compared with common adenocarcinoma [11]. For patients with *BRCA/ATM* and other homologous recombination repair (HRR) gene mutations, poly (ADP-ribose) polymerase (PARP) inhibitors are effective and have recently been accepted [7,8,9,12].

Although a growing body of transition mechanisms to CRPC have been collected, little is known about NM variation’s role in prostate cancer progression, especially in the transition to castration-resistant states. In order to reveal the role of NM proteins in the transition to CRPC, we performed focused a 2D-proteomic analysis of the NM proteins using an androgen-sensitive prostate cancer cell line, LNCaP, and its derivative, LN96, which were established after long-term culturing under androgen-deprived conditions, as a representative model of the transition states, and we reveal that purine-rich element binding protein alpha (PURα) was detected in the NM proteins and differentially expressed between the cell lines [13]. In this review, we will summarize and discuss its role in prostate cancer progression and also review recent knowledge about NM and prostate cancer.

## 2. Nuclear Matrix Proteins and Prostate Cancer

In 1991, Getzenberg and his colleagues analyzed the NM proteins of the Dunning rat normal prostate tissues and its prostate cancer cell lines using 2D-electrophoresis and found that the NM protein composition patterns were different among them [3]. Thereafter, the same group compared NM protein patterns from human normal prostate, benign prostatic hyperplasia (BPH), and prostate cancer extracted from 21 patients. Using the same methods, they identified 14 different protein spots that consistently differ among the various tissues [4]. They identified that one protein called PC-1, with a molecular weight of 56,000 and pI 6.58, could be detected in all NM components of prostate cancer patients but not in those of either normal prostate or BPH patients. Establishing a monoclonal antibody against the molecule, they finally identified that it is a nucleolar phosphoprotein, nucleophosmin [14]. Furthermore, they showed that a specific charged protein, YL-1, of molecular weight 76 kD and pI range 6.0–6.6 was found to be consistently present in 19 of 19 aggressive prostate cancers but was present only in 1 of 10 in the group with good-prognosis prostate cancers [15]. The same NM protein extraction method was applied to 5 patients who received radical cystectomy for bladder cancer (normal prostatic tissues), 17 BPH patients, and 29 patients with radical prostatectomy for prostate cancer, and Alberti et al. found that complexity of the NM protein pattern was observed with an increase in the Gleason score of prostate cancer patients’ tissues compared with normal prostate tissues [16]. They extended this research by analyzing the NM proteins of 75 prostate cancer patients and reported that the expression of three NM proteins was significantly correlated with the risk of biochemical recurrence [17]. One of the NM proteins was identified as heterogenous nuclear ribonucleoprotein K (hnRNP K) [17]. Immunohistochemical staining showed that hnRNP K was overexpressed in prostate cancer tissues both in the cytoplasm and in the nucleus compared to normal prostate tissues [18]. Additionally, the level of hnRNP K in the NM was associated with the degree of prostate cancer differentiation [18]. Using 2D-electrophoresis and Western blotting, hnRNP K presented several isoforms; the one with pI 5.1 was the most differently expressed between non-tumor and PCa tissues [18].

Chromatin is organized into loops dependent upon DNA sequences that tether the chromatin to the NM [19]. These anchor sequences are known as scaffold/matrix attachment regions (S/MARs) [20]. Various proteins, S/MAR binding proteins, are known to interact with S/MARs to facilitate chromatin looping, and DNA looping has been recognized to be crucial for many cellular processes such as DNA replication, transcription, chromatin to chromosome transition, and DNA repair [20]. Thus, it is evident that the interplay between S/MARs and the NM is critical for the organization and function of genetic material. Furthermore, there is no doubt that the alterations of S/MARs and NM interaction could be involved in disease occurrence and progression [21]. Barboro et al. showed that the expression and localization of poly (ADP-ribose) polymerase (PARP) in S/MARs was higher in PC3, an androgen-receptor-negative aggressive prostate cancer cell line, compared with LNCaP, an androgen-receptor-positive differentiated prostate cancer cell line. On the contrary, the expression and localization of special AT-rich sequence-binding protein-1 (SATB1) and its fragments was decreased in S/MARs in PC3 compared to LNCaP [21]. Mao et al. reported that the expression of SATB-1 was higher in prostate cancer patients with metastasis than those without metastasis by immunohistochemistry, and its expression was absent in BPH [22]. Furthermore, SATB1 expression was positively correlated with the Gleason score [22]. Interestingly, analyzing NM proteins bound to S/MAR sequences using human normal prostate and prostate cancer tissues, PARP-1 expression was significantly increased in prostate cancer tissues and its expression was increased in more aggressive prostate cancers (Gleason scores of 8 or 9) compared with a Gleason score of less than 7 in prostate cancer and normal prostate tissues [23]. They inhibited PARP-1 using ABT-888 (Veliparib), a potent PARP-1/2 inhibitor, in vitro using PC3 cells and revealed that it could decrease cell viability, migration, invasion, chromatin loop dimensions, and histone acetylation [23].

## 3. Androgen Receptor and Nuclear Matrix in Prostate Cancer

The androgen receptor (AR) is a component of the NM, and NM-associated AR is strongly associated with cell viability and its transcriptional activity, indicating that its subnuclear localization is critical for cellular activities [24]. A phosphorylated isoform of hnRNP K is localized in the NM of prostate cancer, and the hyperphosphorylation of hnRNP K co-localized with AR in the NM and exerted AR co-activator functions [24]. HnRNP K could bind to the AR mRNA 5′-UTR and also to an AR open-reading frame, resulting in the inhibition of AR translation [25]. Additionally, the expressions of hnRNP K and AR were inversely correlated in localized prostate cancer by immunohistochemistry, but a substantial loss of cytosolic staining of hnRNP K was seen with progression to metastases [25]. A reduction in cytosolic hnRNP K, which could have a role in inhibiting AR translation, might presumably support the AR expression level in a lethal phenotype [25]. Thus, hnRNP K might have different phenotypic properties influencing the AR according to subsets of prostate cancer.

Non-muscle actin is ubiquitously present in eukaryotic cells, has been shown to equilibrate between the nucleus and cytoplasm, and is considered to be one of the component proteins of the NM [26,27]. In order to nucleate actin filaments from a large pool of monomeric actin bound to profilin, eliciting temporal and spatial remodeling of the actin cytoskeleton, which underlies complex cellular functions, actin nucleators are needed [28]. Formins are a diverse protein family discovered in all eukaryotes and are one of the potent nucleators of linear actin filaments. The Dishevelled-associated activator of morphogenesis 2 gene (*DAAM2*) belongs to the formin family. Biallelic pathogenic mutations or variants in *DAAM2* have been described in steroid-resistant nephrotic syndrome [29]. Recently, exome sequencing on 26 androgen insensitive syndrome (AIS) type II genital skin fibroblasts, who are an *AR* mutation-negative group, revealed heterozygous variants in *DAAM2* in two unrelated individuals with partial AIS who were born with hypospadias, a micropenis, no Mullerian structures, and normal plasma testosterone levels [30]. These two *DAAM2* variants, one with a frameshift mutation leading to a premature stop codon and the other with a missense mutation in the regulatory GTPase-binding domain, resulted in a significant reduction in dihydrotestosterone-induced AR activity [30]. And this AR signaling strongly depends on DAAM2 and locally defined and highly dynamic nuclear actin polymerization [30]. Especially, PSA expression in the prostate cancer cell line was sensitive to the pharmacological inhibition of both actin assembly and condensate formation, implicating a potential role for NM actin-dependent AR transcriptional activity in hormone-sensitive prostate cancer progression [30]. Flightless I (FliI) is an actin-binding member of the gelsolin family of actin-remodeling proteins. It consists of an N-terminal leucine-rich repeat domain and C-terminal gelsolin-like domain, and it was previously reported that it could be recruited by hormone-activated nuclear receptors to the promotors of the target genes serving as a coactivator of the nuclear receptor transcription complex [31]. Both in silico and in human prostate cancer immunostaining, the expression of FliI was positively correlated with the overall survival of prostate cancer patients [32]. Moreover, in primary prostate cancers with high AR expression, there was a tendency that the survival of the group with a high expression of FIiI was better than that with low FIiI expression [32]. FIii dissociated from the AR in a ligand-dependent manner and could inhibit AR transactivation and modulate AR cytoplasm/nucleus localization through direct binding to the AR-ligand binding domain [32].

The nuclear matrix protein Scaffold Attachment Factor (SAFB1, *SAFB*) has recently been shown to participate in the higher-order organization of pericentromeric heterochromatin by interacting within major satellite RNAs and forming phase separation. The depletion of SAFB1 leads to a remodeling of 3D genome organization, and genetic variants or the dysregulation of SAFB1 are implicated in various cancers [33]. SAFB1 has a role in prostate cancer, regulating AR as a co-repressor through the epigenic silencing of AR targets, such as PSA [34]. Silencing SAFB1 in LNCaP cells suppressed the expression of the UDP-glucuronosyltransferase family member B15 gene (*UGT2B15*) and the *UGT2B17* gene, which encode proteins that irreversibly inactivates testosterone and dihydrotestosterone [34]. Moreover, a public database revealed a significant decrease in disease-free survival for patients with genomic loss at the *SAFB* locus. A multi-institutional integrative clinical sequencing analysis showed that patients treated either with androgen-deprivation therapy or taxanes expressed a significant decrease in SAFB1 mRNA compared to untreated patients [34,35].

Therefore, NM proteins might have some effects on AR expression or function in prostate cancer, and focusing on that aspect, we should evaluate the relation with NM proteins and AR in prostate cancer progression more deeply.

## 4. Nuclear Matrix Protein, Purine-Rich Element Binding Protein Alpha (PURα) and Prostate Cancer

LN96 cells, which are derivatives of the androgen-receptor-positive, androgen-sensitive cell line LNCaP, were established after long-term culturing under charcoal-stripped fetal bovine serum [36]. In order to discover novel targets for revealing a mechanism of the transition to CRPC, we performed focused proteomic analysis of the composition of the NM proteins [13]. We firstly found that purine-rich element binding protein alpha (PURα), encoded by *PURA*, was detected in the NM composition within both androgen-dependent and -independent prostate cancer cell lines [13]. PURα was decreased in LN96 and PC3 cells compared to LNCaP cells in NM proteins [13].

PURα is a single-stranded DNA/RNA-binding protein that is highly conserved from bacteria to humans [37]. Human PURα is a member of the Pur family of proteins, which consists of four members: Pur-alpha, Pur-beta, and two forms of Pur-gamma [37]. The details of its function were comprehensively described in an excellent review [37]. Briefly summarized, (1) human *PURA* located at chromosome band 5q31 and environmentally induced heterozygous deletions in the lesion where *PURA* is located have been considered as a mechanism of the occurrence of myelodysplastic syndrome [38]; (2) overexpression or microinjection of PURα inhibits the anchorage-independent cell growth of oncogenic cells and blocks proliferation at either G1-S or G2-M checkpoints [39,40]; (3) cell-cycle inhibition may be mediated by the interaction of PURα with Cyclin/Cdk complexes and Rb tumor suppressor proteins [41]; (4) *PURA* knockout mice die soon after birth due to a disorder of brain development; (5) PURα interacts with the HIV-1 protein Tat and plays some role in AIDS; and (6) microdeletions in the *PURA* locus have been implicated in several neurological disorders (PURA syndrome). Contrary to other disorders derived from mutations in a single gene, PURA syndrome patients show high penetrance, meaning almost every reported mutation in the gene leads to symptoms. Thus, its functions have been examined mainly focusing on neurological aspects. And these studies performed using mice showed that PURα is detected predominantly in the cytoplasm [42].

PURα binds to various mRNAs, thereby bearing the potential to regulate a multitude of different cellular processes including interleukin signaling, autophagy, mitochondrial homeostasis, and neurodevelopment. Interestingly, reduced PURα levels decrease the expression of the integral processing (P)-body components LSM14A and DDX6 and affect P-body formation, influencing the formation and composition of the phase-separated RNA processing [43,44].

PURα in nuclear compartments binds single-strand (ss) DNA and ssRNA equally in a similar way. These in vitro results implicate that competition between DNA and RNA could be an important feature for the PURα-dependent regulation of gene expression in cells [42]. The molecular and cellular functions of PURα in neuronal disorders are well-summarized in Molitor et al. [42].

NM proteins were extracted according to the common methods as reported by Fey et al. [45]. Two-dimensional gel electrophoresis of NM proteins derived from LNCaP and LN96 cells showed that a spot of 42 kDa and pI 6.7 was significantly decreased in the NM proteins of LN96 cells [13]. The spot was identified as PURα by mass spectrometric analysis and by two-dimensional gel electrophoresis and subsequent immunoblotting using a commercially available antibody against PURα [13]. And this was also validated by one-dimensional electrophoresis and immunoblotting for NM proteins by the antibody [13]. Furthermore, the mRNA and protein expression of PURα were also decreased in the androgen-independent, androgen-receptor-negative prostate cancer cell lines PC3 and DU145 [13]. Then, we overexpressed PURα in PC3 and DU145 cells and revealed that PURα could significantly reduce cell proliferation both in vitro and in vivo [13,46]. Although, the overexpression or reduced expression of PURα did not change the cell proliferation of LNCaP cells, a common set of genes involved in stress response and cell differentiation was induced in the cells overexpressing PURα [46].

At the same time, Wang et al. found that PURα and hnRNP K bind to the 5′-untranslated regions of the *AR* gene as a novel transcriptional repressor complex [47]. The overexpression of PURα in androgen-independent derivatives of LNCaP cells, AI cells, induced a reduction in both AR expression and its transcriptional activity under androgen-depleted conditions [47]. Additionally, the reduction in PURα by PURα-specific siRNA increased AR expression and its transcriptional activity in LNCaP cells and its effect was more significant than that in AI cells [47]. The reason for this difference was speculated to be that the endogenous expression of PURα was very low in AI cells, so the relative reduction in PURα expression was not significant for the cell [47]. Interestingly, the downregulation of PURα by PURα-specific siRNA induced LNCaP cells to grow under androgen-depleted conditions, indicating that the reduction in PURα expression might have a role in progression to CRPC. Actually, when compared with hormone-naïve and CRPC prostate cancer specimens, CRPC specimens with higher AR expression levels had reduced PURα m-RNA, cytosolic rather than nuclear localization of the PURα protein, and occupancy by PURα of the suppressor element in the 5′-UTR of the human *AR* gene [47].

To understand the gene sets regulated by PURα in prostate cancer cells, the comprehensive gene expression of PC3 and LNCaP when PURα was overexpressed by exogenous introduction was performed in comparison with control cells. Among the increased genes in PURα-overexpressing cells, *ATF3* and *HERPUD 1* have been reported to be regulated by the androgen–androgen receptor (AR) axis. Thus, we speculated that PURα and AR might share common gene pathways. Next, we evaluated the ability of PURα to regulate luciferase expression under control of a 5.3 kb region of the PSA promotor. Using HEK293 cells, which do not express endogenous AR, the exogenous introduction of PURα itself enhanced the reporter activity with or without AR introduction in the absence of androgen stimulation [46]. Since PURα shows a strong preference for binding and unwinding double-strand G-rich DNA repeats, and has the ability to recruit proteins to unwound regions of either DNA and RNA, we would speculate that PURα may recruit proteins such as Sp-1 to enhance the reporter activity of the 5.3 kb region of the PSA promotor without AR or androgen stimulation [48,49,50]. To explore the effect of the overexpression of PURα on endogenous PSA expression under a low androgen concentration (0.1 nM R1881 or 10% fetal bovine serum), LNCaP cells (possess mutant AR) and LAPC4 cells (possessing the wild-type AR) underwent exogenous PURα introduction, resulting in PSA expression being enhanced within both cell lines [46]. These results are compatible with the PSA promotor activity analysis describe above. However, with higher androgen stimulation (1 nM or 10 nM R1881), PURα did not increase or rather decreased AR transcriptional activity [46,47]. Thus, the interaction between PURα and AR might be context-dependent, such as based on the microenvironmental androgen concentration or target cells evaluated (Figure 1).

The evaluation of an estrogen receptor β (ERβ) knockout mouse developed using CRISPR/cas9 technology revealed that nuclear PURα expression was reduced in the ventral prostate of ERβ knockout mouse [51]. Therefore, PURα is an ERβ–induced gene and its expression levels are increased by ERβ agonists in 2- and 6-month-old wild-type mice [51]. But Warner et al. only presented data on the ventral prostate of the ERβ knockout mouse at the age of 18 months, when the detachment and dying of the epithelial cells was apparent due to the reduction in serum androgen concentrations [51]. In the 6-month-old ERβ knockout mouse, many foci of epithelial hyperplasia and intraductal cancer-like lesions with an increase in Ki67 expression were found [51], but they did not show the expression of PURα at that time point. A prostate-specific ERβ knockout mouse remains to be established to provide us with the tools to truly understand the relationship between PURα and ERβ in prostate cells.

## 5. Purine-Rich Element Binding Protein Alpha (PURα) and Cancer

Recently, Yu et al. reported that the expression of the long noncoding RNA (lncRNA) acting gamma 1 pseudogene 25 (AGPG) was increased in endocrine-resistant breast cancer cells by the epigenomic activation of the enhancer [52]. AGPG was upregulated, especially in patients with luminal B subtype breast cancers, and a high AGPG expression was associated with poor survival in estrogen receptor (ER)-positive breast cancers [52]. Interestingly, they identified that AGPG has a potential binding interaction with PURα in the nucleus [52]. Previously, PURα has been shown to interfere with the transcriptional activation of the S-phase-specific gene dihydrofolate reductase (DHFR) by interacting with E2F1 and preventing its association with the E2F1 elements located within the DHFR promoter [53]. The same as in this report, PURα and E2F1 were co-localized in the nuclear fraction of breast cancer cells, and AGPG disrupted the formation of the PURα/E2F1 complex to activate E2F signaling, facilitating ER-positive breast cancer cell proliferation and endocrine resistance [52]. Moreover, a clinically relevant transcriptome database showed that AGPG and PURα positively and negatively enriched 195 E2Fs targets, respectively. On the other hand, most E2F targets were positively correlated with AGPG expression and the most of them were negatively correlated with PURα expression [52]. Thus, in breast cancer cells, PURα can function as a tumor suppressor as in prostate cancer cells described above.

On the contrary, Gao et al. revealed that the expression of PURα enhanced esophageal squamous carcinoma (ESCC) tumor growth both in vitro and in vivo [54]. They also showed that PURα promoted ESCC cell migration, invasion in vitro, and also metastasis in vivo [54]. Using human ESCC patients’ tissue, the expression of PURα positively correlated with lymph node metastasis and the American Joint Committee on Cancer stages [54]. Furthermore, conducting transcriptome profiling after PURα overexpression and knockdown in ESCC cell lines by RNA sequencing, they showed that PURα plays a role in the epithelial–mesenchymal transition (EMT) pathway, which is a critical event in tumor invasion and progression through indirectly regulating Snail2 expression [54]. The same group further revealed that the expression level of cytoplasmic PURα in ESCC tumor tissues was significantly higher than in adjacent epithelia [55]. Since immunofluorescence staining showed that PURα was significantly localized in cytoplasm as granules or accumulated around the nucleus in non-granules, and co-localized with G3BP1, a well-known cytoplasmic stress granule marker, cytoplasmic PURα is a component of stress granules in ESCC cells [55]. Exploring its function, PURα directly binds mRNA with a strong preference for UG-/U-rich motifs of 3′-UTRs and inhibits *IGFBP3* protein expression, regulating its mRNA translation [55]. Cytoplasmic PURα recruits translation initiation factors to modulate protein expression, consistent with the characteristics of cytoplasmic stress granules [55]. Importantly, immunohistochemical analysis revealed that ESCC tissue with higher expression levels of cytoplasmic PURα had a lower expression of IGFBP3, supporting that PURα inhibits the mRNA translation of IGFBP3 [55]. Since IGFBP3 exhibits antitumor properties in ESCC cells and IGFBP3 expression was significantly decreased in ESCC tissues, cytoplasmic PURα may mediate ESCC progression by regulating mRNA translation [55]. Therefore, at least in ESCC cells, PURα accelerates the progression of tumors.

Considering to the multiple functions of PURα, including DNA transcription, duplication, and mRNA transport, the heterogenous functions among different types of cells might result in different effects of PURα on the progression of different cancers [38,40] (Table 1) (Figure 2).

## 6. Conclusions

In summary, PURα is a nuclear matrix protein identified in prostate cancer cells. Nuclear PURα can regulate AR activity depending on cellular contents, but under sufficient androgen stimulation it might inhibit AR transactivity. Additionally, according to the expression analysis, a decrease in the nuclear expression of PURα in mCRPC specimens may indicate its tumor suppressive role in the progression to CRPC in prostate cancer. Since PURα seems to have oncogenic roles in ESCC cells, which is the opposite to the role seen in prostate cancer, the role of PURα, especially as a component of the NM, remains to be elucidated and further investigation needs to be performed.

## Figures and Tables

**Figure 1 ijms-25-06911-f001:**
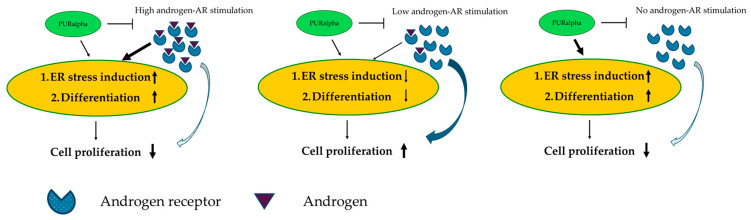
Diagram of the suspected role of PURα in prostate cancer cells. At a high androgen concentration, androgen/androgen receptor (AR) stimulates endoplasmic reticulum stress and cell differentiation but reduced cell proliferation in prostate cancer cells. At a low androgen stimulation, androgen/AR induces cell proliferation in prostate cancer cells. Without androgen, PURα induces endoplasmic reticulum stress and cell differentiation, resulting in a reduction in prostate cancer cell proliferation. On the other hand, AR may be negatively regulated by PURα but its involvement might be relatively low [46,47].

**Figure 2 ijms-25-06911-f002:**
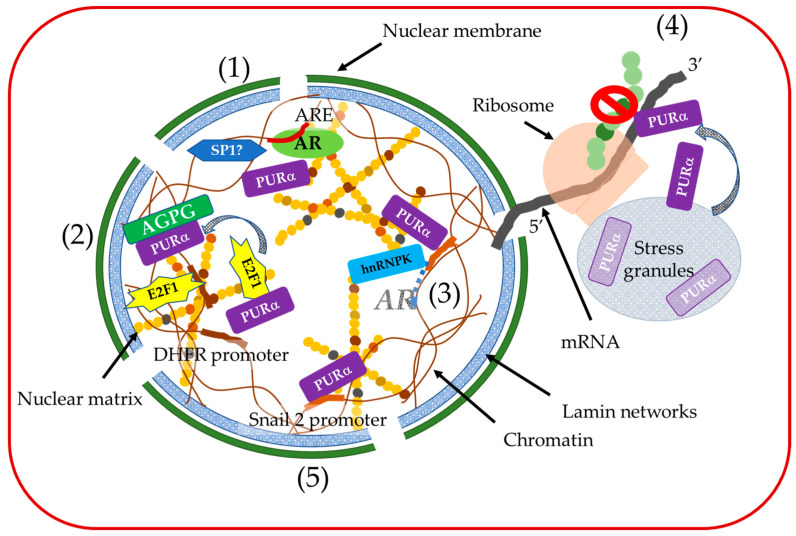
Diagram of the suspected role of PURα in cancer cells. (1) PURα upregulates or downregulates the androgen response element (ARE) together with other transcriptional factors such as Sp-1 depending on the androgen concentration [13,46]. (2) PURα interferes with the transcriptional activation of the S-phase-specific gene dihydrofolate reductase (DHFR) by interacting with E2F1 and preventing its association with the E2F1 elements located within the DHFR promoter. Acting gamma 1 pseudogene 25 (AGPG) disrupts the PURα/E2F1 interaction, activating E2F1 signaling pathways [52,53]. (3) PURα and hnRNP K bind to the 5′-untranslated regions of the *AR* gene as a novel transcriptional repressor complex [47]. (4) PURα directly binds mRNA with a strong preference for the UG-/U-rich motifs of 3′-UTRs and inhibits *IGFBP3* protein expression (stop mark) [55]. (5) PURα binds to the Snail2 promoter, resulting in the downregulation of E-cadherin and accelerating cell proliferation and migration [54].

**Table 1 ijms-25-06911-t001:** Summary of PURα and its role in cancers.

Authors	Year of Publication	Target Cancer	Role of PURα
Wang et al. [47]	2008	Prostate cancer	Induces the attenuation of androgen receptor expression and transcriptional activity.
Yu et al. [52]	2023	Breast cancer	Forms a complex with E2F1 and its complex is disrupted by long noncoding RNA, acting gamma 1 pseudogene 25.
Gao et al. [54]	2021	Esophageal cancer	Induces epithelial–mesenchymal transition through the indirect regulation of Snail2.
Tian et al. [55]	2022	Esophageal cancer	Cytoplasmic PURα, a component of stress granules, inhibits the mRNA translation of IGFBP3.
Inoue et al. [13,46]	2008, 2009	Prostate cancer	Reduces the cell proliferation of prostate cancer cells through the stimulation of endoplasmic stress and cell differentiation. Additionally, PURα enhances PSA promoter activity without androgen stimulation.

## Data Availability

Raw data are published in the manuscript.

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
