# Peer review of "Purine-Rich Element Binding Protein Alpha, a Nuclear Matrix Protein, Has a Role in Prostate Cancer Progression"

_ijms, 2024, doi:10.3390/ijms25136911_

Round 1

Reviewer 1 Report

Comments and Suggestions for Authors

In this manuscript the authors review the potential role of purine-rich element binding protein alpha in the nuclear matrix proteins and its expression difference between androgen-sensitive and androgen-independent prostate cancer cell lines. They summary various NM proteins with molecular weight and PI in prostate cancer and the relationship between S/MAR of chromatin anchor sequences and NM proteins. In addition, the effects of NM proteins on androgen receptor expression and function in prostate cancer were also reviewed. The interaction between PURα and AR was also explained by context-dependent mechanism. Finally, the expression of PURα in other cancers such as breast cancer and esophageal cancer were also summarized. This review provides sufficient depth in discussion and enough details for examples. I would suggest the author add a list of abbreviation. In addition, I was wondering if there is any cancer treatment study that target to PURα.

Author Response

In this manuscript the authors review the potential role of purine-rich element binding protein alpha in the nuclear matrix proteins and its expression difference between androgen-sensitive and androgen-independent prostate cancer cell lines. They summary various NM proteins with molecular weight and PI in prostate cancer and the relationship between S/MAR of chromatin anchor sequences and NM proteins. In addition, the effects of NM proteins on androgen receptor expression and function in prostate cancer were also reviewed. The interaction between PURα and AR was also explained by context-dependent mechanism. Finally, the expression of PURα in other cancers such as breast cancer and esophageal cancer were also summarized. This review provides sufficient depth in discussion and enough details for examples. I would suggest the author add a list of abbreviation. In addition, I was wondering if there is any cancer treatment study that target to PURα.

Response to the reviewer:

Thank you for your thoughtful comments.  We have added a list of abbreviation for the readers of the journal to easily understand this manuscript. 

As far as we know, there is no trial targeting PURα for cancer treatment.

Reviewer 2 Report

Comments and Suggestions for Authors

The Authors made a great effort in characterizing the Purine-rich element binding protein alpha in cancer progression. I believe the paper would be better if the Authors:

1. Provide a molecular scheme of cancer cell with the localization and cellular function of Purine-rich element binding protein alpha.

2. Focus more on the localization of the protein and its function in cancer-associated molecules transport. 

3. Provide some clinical-linked data concerning the potential application of the knowledge about the Purine-rich element binding protein alpha in cancer treatment. 

4. Comprehensively describe the molecular pathways (with schemes!) in which Purine-rich element binding protein alpha is involved. 

Otherwise a nice paper!

Author Response

The Authors made a great effort in characterizing the Purine-rich element binding protein alpha in cancer progression. I believe the paper would be better if the Authors:

  1. Provide a molecular scheme of cancer cell with the localization and cellular function of Purine-rich element binding protein alpha.

Response to the reviewer:

We have added a scheme of intracellular functions of PURα with its localization focusing cancer cells in Figure 2.  Additionally, we also added some description about the localization and cellular function of PURa especially focusing neurologic disorders on page 5 in line 221-236.

  1. Focus more on the localization of the protein and its function in cancer-associated molecules transport. 

Response to the reviewer:

There are few reports focusing on roles of PURa in molecular transport especially in cancer cells. Instead we added a scheme of intracellular functions of PURα with its localization focusing cancer cells in Figure 2.

  1. Provide some clinical-linked data concerning the potential application of the knowledge about the Purine-rich element binding protein alpha in cancer treatment. 

Response to the reviewer:

There is no report of clinical application of the knowledge about PURa in cancer treatment. 

  1. Comprehensively describe the molecular pathways (with schemes!) in which Purine-rich element binding protein alpha is involved. 

Response to the reviewer:

We have added a scheme of intracellular functions of PURα with its localization in Figure 2.

Round 2

Reviewer 2 Report

Comments and Suggestions for Authors

The Authors sufficiently addressed all my issues and now I belief the manuscript may be accepted for publication!